# Retrospective Analysis of Injuries and Hospitalizations of Patients Following the 2009 Earthquake of L’Aquila City

**DOI:** 10.3390/ijerph16101675

**Published:** 2019-05-14

**Authors:** Jacopo Del Papa, Pierpaolo Vittorini, Francesco D’Aloisio, Mario Muselli, Anna Rita Giuliani, Alfonso Mascitelli, Leila Fabiani

**Affiliations:** 1Postgraduate Schools of Hygiene and Public Health—Department of Life, Health and Environmental Sciences, University of L’Aquila, P.le S. Tommasi, 1, 67100 Coppito, L’Aquila, Italy; francescodalo86@gmail.com (F.D.); mario.muselli@hotmail.it (M.M.); 2Department of Life, Health and Environmental Sciences, University of L’Aquila, P.le S. Tommasi, 1, 67100 Coppito, L’Aquila, Italy; pierpaolo.vittorini@cc.univaq.it (P.V.); annarita.giuliani@cc.univaq.it (A.R.G.); leila.fabiani@univaq.it (L.F.); 3Regional Health Agency of Abruzzo Region (Italy)—Via Attilio Monti, 9, 65127 Pescara PE, Italy; direzionegenerale@asrabruzzo.it

**Keywords:** injury, earthquake, natural disaster, trauma, fractures, hospital discharge records, ICD-9-CM, disaster medicine, Italy, L’Aquila

## Abstract

The aim of this study was to investigate the injury patterns and the hospitalizations of patients who were admitted to hospital following the 2009 earthquake in the city of L’Aquila, Central Italy. To the best of our knowledge, this is the first study to analyze the patterns of earthquake-related injuries in Italy. We reviewed the hospital discharge data of 171 patients admitted to hospital within the following 96 h from the mainshock. This is an observational and descriptive study: We controlled for variables such as patient demographics, primary and secondary ICD-9-CM (International Classification of Diseases) diagnosis codes in order to identify the multiple injured patients, main type of injury that resulted in the hospital admission, discharge disposition, and average length of stay (LOS). Seventy-three percent of the 171 patients were admitted to hospital on the first day. Multiple injuries accounted for 52% of all trauma admissions, with a female to male ratio of 63% versus 37%. The most common type of injuries involved bone fractures (46.8%), while lower extremities were the most frequently affected sites (38.75%). The average LOS was 12.11 days. This study allows the evaluation of the impact of earthquake-related injuries in relation both to the health needs of the victims and to the use of the health care resources and assistance.

## 1. Introduction

In the last 100 years, earthquakes have claimed millions of lives and injured thousands of people [1]. An effective and well-conceived emergency management system designed to reduce the mortality and morbidity associated with earthquakes is crucial for all the health care workers involved in disaster management [2,3]. Thus, the analysis and study of traumatic injuries in patients affected by major disasters is fundamental to assess the patterns of disaster-related injuries, in order both to enhance health care delivery in the event of a major disaster and to train emergency medical responders [4]. Italy lies on the boundary where the African and Eurasian tectonic plates converge. The movement of these two big plates colliding results in the deformation and high amounts of energy that are occasionally released in earthquakes of different magnitude. Since 1900 to date, there have been 30 large earthquakes (Mw ≥ 5.8), some of which were devastating. The most powerful earthquakes recorded in the last few years hit Abruzzo on 6 April 2009, Mw = 6.1, Emilia-Romagna on 20 May 2012, Mw = 5.8 [5], and Central Italy in 2016 and 2017.

The earthquake that struck the city of L’Aquila in 2009 consisted in a series of foreshocks, which began in December 2008 and ended in 2012 [6], with epicenters in the L’Aquila city and province. The mainshock struck at 03:32 a.m., Central European Time, on 6 April 2009 [7], with a moment magnitude (MMS) of 6.3 (5.8 or 5.9 as measured by the local magnitude scale), and with geographical coordinates of the epicenter 42°20′51.36″ N 13°22′48.4″ E, located in the hamlet of Colle Miruci, in Roio (L’Aquila), and was widely felt with varying degrees of intensity in a large part of Central Italy [8]. Based on the local magnitude scale rating (the so-called Richter scale, which however does not provide accurate estimates for large magnitude earthquakes [9]), the seismographs measured a magnitude 5.9 ML tremor, thus resulting in a moderate seismic event if compared to the maximum values that can be reached on this seismic magnitude scale [10]. Indeed, the Peak Ground Acceleration [11], which is the maximum acceleration of an earthquake on the ground surface, during the 6 April 2009 quake, reached values of up to 0.68 g, a value that can be theoretically assigned to 7.2–7.4 magnitude earthquakes [12]. Based on the Mercalli intensity scale, which rates the observed structural damage of earthquakes, the initial estimate by the National Institute of Geophysics and Volcanology (INGV—Istituto Nazionale di Geofisica e Vulcanologia) ranged between the VIII and IX intensity level. The mainshock was followed by 256 aftershocks over the course of the next 48 h, of which more than 150 occurred on Tuesday, 7 April, with 56 aftershocks larger than magnitude 3.0 ML. Three events with magnitude greater than 5.0 occurred on 6, 7 and 9 April. The earthquake caused 309 casualties [13], more than 1600 people were injured, 70,000 displaced from their homes, and the cost of the damage was estimated to be more than 10 billion Euros.

During the L’Aquila earthquake, the treatment of injured patients was delayed due to collapsed roadways and the severe structural damages of the main local hospital, the ‘Ospedale San Salvatore’ (following the mainshock, 90% of the hospital’s facilities were severely damaged) [14,15]. Generally, earthquakes cause a higher number of injuries and morbidity than mortality, and the most common earthquake injury pattern involves the musculoskeletal system [16].

In their systematic literature review on the impact of earthquakes on human health, Doocy et al. reported that soft tissue injuries (including lacerations and contusions), fractures (in particular, limbs), and crush injury to head, thorax and abdomen caused by buildings collapsing, are the most common types of injury [17]. They can abruptly result in death by asphyxiation, bleeding and acute kidney injury (due to crush syndrome) [18,19]. A body of studies has focused on the different patterns of fractures, as they represent half of the injury-related hospital admissions and a major reason of long-term disabilities [12,20]. Unlike other natural disasters, earthquakes are unpredictable, thus often causing catastrophic and uncontrollable events [21]. However, to date, the patterns of earthquake-related injuries, and in particular the trauma caused by the earthquakes that struck Italy in the past years, have not yet been extensively studied [16,22].

This study aimed to analyze the trauma admissions within the first 96 h after the mainshock, with a focus on the injury patterns, the features of earthquake-related injury admissions, the in-hospital mortality rates, and other related health outcomes. Although an analysis of the patterns of injury admissions following the L’Aquila earthquake has not yet been conducted, it may be relevant to enhance the assistance capability in high-income countries with high-standard health care services, and to provide recommendations to improve the Italian health care system during and after major disasters. On an international level, research studies that specifically target the injury patterns following high-magnitude earthquakes have not yet been conducted at a high level of scientific standards, due to the scarcity and poor quality of the data collected. As reported by Bortolin et al., the characterization of a specific disaster-related injury pattern is crucial to build an effective disaster preparedness and response system, including the development of guidelines and the definition of standards-of-care (SOC) specific to any, natural or man-made, mass casualty event [4].

## 2. Materials and Methods

This is an observational and descriptive study based on current data. We reviewed the hospital discharge forms of 171 patients involved in the L’Aquila city earthquake, and from the so-called “seismic crater” geographic area (arbitrarily comprising of the city of L’Aquila and 41 smaller municipalities struck by the 03:32 a.m. earthquake and by the aftershocks), who were admitted to 30 regional and extra-regional hospitals [8]. We conducted a retrospective analysis of all inpatient hospitalizations recorded in the large discharge database of the Abruzzo region [23]. We excluded from the study outpatient surgeries, rehabilitation hospital stays, and long-stay hospital patients. Moreover, we did not report all the injury-related admissions to the ‘Ares Marche’ field hospital, as no hospital discharge forms were released [24]. We used the primary diagnosis codes from the chapter titled “Injury and Poisoning” with ICD-9-CM codes from 800–959 of the Italian version of the ICD-9-CM “International Classification of Diseases—9th revision—Clinical Modification” 2007 (although they are part of the same chapter, the ICD-9-CM diagnosis codes in the range 960–999 have not been taken into account as they were not reported as a cause of hospitalization and they are also related to poisoning). Data include injury-related hospital admissions, age, gender, type of discharge, in-hospital mortality and average length of stay (LOS). We also analyzed the secondary diagnoses and whether they reported any injury code (800–959 ICD9-CM) to identify multiple-injured patients. We compared the inpatient admissions and the types of discharge from 6 to 9 April 2009 with all the years following the earthquake (2010–2016) in order to determine any difference between the earthquake-related admissions and discharges and those following the event.

### 2.1. Statistics

Data on the admissions of the population from the seismic crater area (L’Aquila city and province), were collected from file A of the HDRs (hospital discharge records), and they refer to the time period from 6 April 2009 to 9 April 2009. The hospital discharge forms were retrieved from the information and health statistics management service of the Abruzzo Region Health System. For the analysis of data expressed in terms of frequencies, we used the **χ**^2^-Test and the Fisher Test. To perform the analyses of numerical data, such as the LOS, we used the T-Test or the Wilcoxon test if the normality requirement was not respected. Furthermore, we used the Cramér’s V test to measure the association between two nominal variables. For the statistical analysis, we used software R, version 3.4.3 (The R Foundation, Vienna, Austria) for Windows.

### 2.2. Seismic Crater Area

It is a geographical subset of the province of L’Aquila, which comprises of the city of L’Aquila, with a population of 72,696 inhabitants, and 41 smaller municipalities (a total of 38,874 inhabitants) [8]. Such selection was based on the earthquake-related structural damages. Although some municipalities of the neighboring provinces were struck by the earthquake, they were not included in the study as they are regulated by different local health authorities with different hospital admission policies.

### 2.3. Ethical Considerations

This study was approved by all of the participating researchers and the Ethical Committee (IRB: Internal review board) of the University of L’Aquila on 16 January 2018 and registered on 1 February 2018 with no. 4904.

### 2.4. Methodological Limitations

The use of HDRs in epidemiology has strengths, limitations and may introduce biases. The most important strengths are that data already exist, are large and are collected independently from research purposes. The limitations are that data are pre-collected by non-researchers, with a low or unknown quality. Biases may be also introduced, like misclassification as the result of unclear or erroneous clinical documentation, or like the fact that expensive medical procedures are usually documented better than those less costly.

In the few days following the earthquake under investigation, the Abruzzo hospitals and in particular the L’Aquila hospital, focused on the management of the emergencies related to the event. All other emergencies were redirected to other hospitals like the ‘Ares Marche’ field hospital. However, there is no written note on the HDRs that may guarantee that all admissions are directly due to the earthquake, and—for privacy reasons—we could not contact the patients involved in the study for a confirmation.

## 3. Results

### 3.1. Analysis of the All-Cause Hospital Admissions

From 6 to 9 April 2009, injury-related hospitalizations in the seismic crater area accounted for 30.4% of all admissions, with 171 hospitalized patients, ranging from 48.5% on 6 April with 125 inpatient admissions in the day and as leading cause of hospitalization, to 12% on 9 April with 11 hospitalized patients and as third cause of admission (Table 1).

### 3.2. Injury Pattern

On 6 April, 125 patients were admitted to hospitals (Table 2) accounting for 73.1% of all hospital admissions. From the second to the fourth day after the earthquake, admission percentages ranged from 13.5% on 7 April, to 7% on 8 April, and to 6.4% on 9 April. The most frequent traumatic injuries were “fractures” (80 patients, 46.8%), followed by “internal injury of thorax, abdomen and pelvis” (25 patients, 14.6%). “Intracranial injuries, excluding those with skull fractures” (21 patients, 12.3%) were the third cause of hospitalization. Further causes were “traumatic complications and unspecified injuries” (13 patients, 7.6%), “contusion with intact skin surface” (10 patients, 5.9%), “crushing injuries” (5 patients, 2.9%) and “late effects of injuries, poisonings, toxic effects, and other external causes” (5 patients, 2.9%).

Women had almost a twofold higher rate of injury than men, with frequency rates of 63.7% and 36.3%, respectively. Fractures (ICD-9: 820–829), internal injury of thorax, abdomen, and pelvis (ICD-9: 860–869), contusion with intact skin surface (ICD-9: 920–924), traumatic complications and unspecified injuries (ICD-9: 958–959), and late effects of injuries and other external causes (ICD-9: 905–909) were more frequent in women, while intracranial injuries, excluding those with skull fracture (ICD9-CM: 850–854), were more common among males.

### 3.3. Comparison of Injury-Related Hospitalizations between 6–9 April 2009 and 2010–2016

We then compared the overall frequencies sorted by sex and the average LOS for the injury-related admissions from 6 to 9 April 2009 among the population in the seismic crater with the number of hospitalizations among the overall Abruzzo population during 2010–2016 (Table 3 and Table 4).

Comparisons with the 2010–2016 hospital admissions showed that women reported a higher number of fractures (+24%), and hospitalizations (+24%) than males (Table 3), with a *p*-value = 0.045 *. In 2010–2016, the ratio of male to female hospitalizations was similar in the fractures, with a (M 47% vs. F 53%), and in the overall injury-related hospitalizations with a (M 48% vs. F 52%), compared to the admissions from 6 to 9 April 2009, where the fracture-related admission ratio was (M 35% vs. F 65%), and the overall injury hospitalizations had a ratio of (M 36% vs. F 64%), with a *p*-value = 0.002 *.

As displayed in Table 4, fracture-related admissions reported a similar percentage in terms of total frequencies between 2009 (46.8%) and 2010–2016 (45.6%); the “internal injury of thorax, abdomen, and pelvis” had a higher frequency in the 2010–2016 period (19.7%) vs. 2009 (14.6%) = +5.1%. The “intracranial injury, excluding those with skull fracture” with 2009 (12.3%) vs. 2010–2016 (7.2%) = +5.1%, the “contusion with intact skin surface” with 2009 (5.8%) vs. 2010–2016 (2.5%) = +3.3%, and “certain traumatic complications and unspecified injuries” with 2009 (7.6%) vs. 2010–2016 (3.5%) = +4.1%, had a higher frequency in the subjects hospitalized following the earthquake. Conversely, the “late effects of injuries, poisonings, toxic effects, and other external causes” with 2009 (2.9%) vs. 2010–2016 (10.7%) = −7.8%, and the “sprains and strains of joints and adjacent muscles” with 2009 (2.3%) vs. 2010–2016 (5.6%) = −3.3%, had higher frequency rates in 2010–2016. In 2009, the average LOS was 13.1 days, ranging from a minimum of 1.5 days for the “open wounds of head, neck, and trunk” to a maximum of 26.8 days for the “late effects of injuries, poisonings, toxic effects, and other external causes”. Results indicated that the average LOS was higher in almost all injury-related admissions during the earthquake days than in 2010–2016 (particularly for fractures, with a difference of +2.9 days and *p*-value= 0.08), except for the “open wounds of head, neck, and trunk” and “sprains and strains of joints and adjacent muscles”, which had a longer LOS in 2010–2016. A difference of +2.2 days in the average LOS for all hospitalizations is observed between 6–9 April 2009 and 2010–2016, *p*-value= 0.03 *.

### 3.4. Analysis of the Fractures

As mentioned above, fractures were the most prevalent injury pattern. Figure 1a,b show the total admission rates by anatomic site (Figure 1a), and the distribution of the hospitalizations per day by anatomic site (Figure 1b).

The most commonly injured anatomic sites within the four days after the mainshock were lower extremities (38.75%), spine and trunk (32.5%), and upper limbs (21.25%). Skull fractures accounted only for 7.5%. On 6 April 2009 (Figure 1b), fractures of spine and trunk (ICD-9: 805–809; 22 hospitalizations) were the major reasons for admission to hospital, followed by injuries to lower limbs (ICD-9: 820–829; 22 hospitalizations). Conversely, on 7 April, fractures in the lower limbs (8 patients) were the main cause for hospital admission, whereas on the third and fourth day, fractures in the upper limbs (ICD-9: 810–819; 22 hospitalizations) represented the first cause of hospitalization.

Table 5 analyzes the hospitalizations reported in Figure 1a,b, which illustrate the distribution of fractures by anatomic site, sex, and average LOS. As shown in Table 5, the most frequently injured sites were lower extremities (38.9%, ICD-9: 820–829); specifically, the tibia/fibula (ICD-9: 823) in 10 cases (12.5%), femur in 15 cases (18.8%), followed by “fractures of spine and trunk” (32.5%, ICD-9: 805–809). In this ICD-9 code subset, the most common fracture location was the vertebral column reported in 20 patients (25%); specifically, closed fractures of lumbar vertebra without mention of spinal cord lesion (ICD-9: 805.4) in 11 cases, of the thoracic vertebra in 6 (no.3 ICD-9: 805.2, no.2 ICD-9: 806.35, no.1 ICD-9: 806.36), and of the cervical vertebra in the remaining 3 cases (no.1 ICD-9: 805.8, no.1 ICD-9: 806.06, no.1 ICD-9: 805.01), followed by upper extremity fractures (21.1%, ICD-9: 810-819), including fractures of radius and ulna (ICD-9: 813) in 8 cases (10%), and fracture of humerus (ICD-9: 812) in 7 cases (8.8%).

Additionally, Table 5 highlights that spinal fractures were more prevalent in women (F:M = 3:1), along with fractures to humerus (F:M = 6:1), radius and ulna (F:M = 3:1), and femur (F:M = 4:1). The average length LOS (Table 5) was particularly high for the hospitalizations associated with vertebral column fractures (18.4 days), and with the fractures to femur (14.93 days), which were also the most common type of injuries within the four days following the mainshock. Admissions to hospital due to fractures of the pelvic girdle (34 days), of the orbit (22 days), and of the phalanges of foot (22 days), experienced the longest average LOS.

### 3.5. Gender and Age Pattern

The reviewed data on the patients from the seismic crater area show a female to male ratio of 1.75:1 (F = 64% vs. M = 36%), with an age range of 0–104 years (Figure 2). The mean age of all patients admitted to the hospital for traumatic injuries was 59.5 ± 23.1 years. Of these, 62 patients (36%) were men, with an average age of 54.9 ± 24.6 years, and 109 patients (64%) were women with an average age of 62 ± 21.9 years. The most represented age group is that of patients older than 60 years (57%, 97 patients), followed by the 17–59 years old group (39%, 67 patients). The pediatric population admitted to hospital with an injury diagnosis accounted for the smallest group (4%, 7 patients). Overall, 97 patients (57%), almost half of the population of this study, were older than 60 years at the time of admission to the hospital. The age differences by gender are not statistically significant (Fisher exact test, *p*-value = 0.19) and a weak association is shown (Cramér’s V = 0.14).

In Table 6, we compared the frequencies sorted by age for the injury-related admissions from 6 to 9 April 2009 among the population in the seismic crater with the number of hospitalizations among the overall Abruzzo population during the 2010–2016 period. In the comparisons between 2009 and 2010-2016, the age group “60 and older” had higher frequencies in the fracture-related admissions (60% vs. 52.2% = +8%) in the “internal injury of thorax, abdomen, and pelvis” (56% vs. 47.2%= +9%) and “intracranial injury” (72.4% vs. 53.3%= +18%). The analysis of *p*-value shows no statistical significance.

### 3.6. Multiple Trauma Patients

The review of the hospital discharge forms highlighted that on 6 April, 59% of the patients were admitted to hospital with secondary ICD-9-CM diagnosis codes in the range 800–959 (Table 7). In the days after 6 April, the percentage of multiple injured patients admitted to the hospital was never below 30%. In total, 90 patients (52% of the total) were admitted to hospital between 6 and 9 April with a multiple trauma diagnosis. The rate of such hospitalizations showed a peak among females, accounting for 57 patients (63%), while the most affected age groups were patients older than 60 years (66%), followed by 17–59 years (28%).

### 3.7. Discharge Pattern

The following frequencies in the discharge patterns in the days following the earthquake were observed:

As summarized in Table 8, of the 171 patients admitted from 6 to 9 April 2009, 3 (1.8%) died; pattern no.2 (discharge to patient’s home) reported the highest rate of inpatient discharge accounting for 70.8% (121 patients), followed by pattern no. 6 (transfer to another, public or private, acute care facility) with 7.6% (14 patients). Furthermore, based on pattern no.8, 4.7% (8 patients) of inpatients were transferred to another rehabilitation facility. The hospital discharge forms do not report whether patients suffered from any permanent impairment as a result of an injury, or the duration of the rehabilitation.

A further analysis compared the percentages of the discharge patterns for the patients admitted within the days following the earthquake with the discharge rates for all injured hospitalized patients in 2010–2016 across the seismic crater area.

Comparison of in-hospital mortality rate between the earthquake days and 2010–2016 (1.8% vs. 1.3%) did not show any noteworthy difference. The discharge to home pattern (no.2) among the inpatients admitted within the days following the earthquake was approximately 15% lower than 2010–2016 period. Conversely, the rate of discharge pattern no.6 (transfer to another, public or private, acute care facility) was higher among the patients admitted from 6 to 9 April (8.2% vs. 1.7%); the rate for pattern no.8 were also higher in the days following the earthquake. The inferential analysis regarding the discharge patterns shows no statistically significant differences in terms of gender, statistically significant differences in terms of period.

### 3.8. Analysis of the Average Length of Stay

As shown in Table 9 the average LOS for all admissions was longer than the overall stays for injury admissions (16.6 days vs. 12.1 days). Average LOS for patients admitted on 6 April (13.2 days) was longer than the admissions within the following three days (7 April: 10.8 days; 8 April: 7.5 days; 9 April: 7.3 days); female inpatients also had a longer average hospital stay than males (12.4 days vs. 11.5 days).

Conversely, Table 10 highlights among the multiple trauma patients, males had a longer average stay than females (12.3 days vs. 11.2 days). The average LOS among multiple injured patients was slightly shorter than the hospital stay of all injury-related patients (11.2 days vs. 12.1 days).

## 4. Discussion

On 6 April 2009, approximately 50% of patients (Table 1) were admitted to hospital with injury as principal diagnosis. This data highlights the importance of exploring injury patterns from major natural disasters, such as earthquakes, and the high number of medical response resources deployed to provide assistance to patients with similar needs. As in the Bam earthquake (Iran) [25], fractures were the most common type of injury, with lower extremities being the major site of injury (ICD-9: 820–829), as also reported in the earthquakes that hit Yushu county (China) in 2010 [20] and West Sumatra (Indonesia) in 2009 [26]. Our review of the fractures classified by their anatomic locations show that the frequency rates and the average LOSs were higher for the fracture of lower limbs with ICD-9: 820–829 (38.8% and 11.5 days of average length of stay), and for fractures of spine and trunk with ICD-9: 805-–809 (32.5% and 18.4 days of average length of stay). Specifically, fractures of vertebral column and of femur were the most frequent injuries and with the longest LOS. Females sustained a higher rate of injuries than males (F:68.75% vs. M:31.25%—Table 5).

Earthquake-related injuries were mostly caused by the collapse of buildings, like in the 2013 Lushan earthquake (China) [21]. This finding is particularly relevant as it has been demonstrated that during an earthquake the incidence of certain injuries and the anatomic location of the fracture are related to the body posture and the activities the victims were performing when the disaster occurred [20,27]. During the day, victims are generally standing or sitting at the time of the disaster, therefore the most frequent observed fractures will involve the vertebral column. Conversely, if the disaster occurs during the night, as in the L’Aquila earthquake (03:32 a.m.), but also in Northridge, California (04:30 a.m. local time), Bam (05:26 a.m. local time), and in Hanshin-Awaji (05:46 a.m. local time), patients are lying in supine or lateral positions at the time of the earthquake, thus the lower extremities (including the pelvis) and the thoracic cage (both fractures and injuries) will be the most common injured sites [28]. As it occurred in the 2011 Van earthquake (Turkey), also in L’Aquila more than 70% of the patients were admitted to hospitals on the first day [1] through the deployment of regional and extra-regional rescue teams from the urgent and emergency care network. As highlighted by Schulz et al. [29], the greatest demand for patient care occurs during the first 24 to 48 h after the disaster (maximum within 4 days after the mainshock). Indeed, the other 46 patients (26.90%) were admitted within the three days following 6 April. After the fifth day, the injury-related emergencies rapidly decreased, thus proving that immediate provision of emergency lifesaving care is fundamental [30], as it occurred in L’Aquila as a result of a joint relief effort by the urgent care and medical emergency services.

Further, the review of the medical records of patients admitted with an injury diagnosis highlighted a recurring injury pattern in the L’Aquila city earthquake: It outlined how females had almost a two times higher incidence of injuries than males (Table 3, all injuries, 36% vs. 64%), whereas injuries in the period 2010–2016 occur in males and females with similar percentages (48% vs. 52%), which are inline—according to the National census data—with the gender distribution in the Abruzzo population (48.6% males and 51.4% females). A breakdown in terms of age groups actually shows a higher frequency of females than males in almost all age groups. However, the association is quite weak (V = 0.14) and therefore the age difference should be considered only a part of the explanation of such a high proportion. This point is also well substantiated by the literature review; most authors claim that women are more at risk for injury and death in earthquakes [25,31,32]. Different studies have pointed out that there may be different approaches between males and females in dealing with a sudden catastrophic event, e.g., due to both traditional stereotyping or cognitive appraisal of threat, and our study seems to support this finding [33,34,35].

This point concurs with our results that, in the L’Aquila earthquake, the most common fractures (femur and vertebral column) were accounted for by females, who also had longer median LOS (with a higher severity score), and half of the inpatients were older than 60 years at the time of admission to the hospital.

The paediatric population constituted only 4% of all inpatient admissions, differently from other earthquakes, such as the Kocaeli earthquake of 1999, in Turkey [36], where the proportion of children who reported injuries was significantly higher.

Much of the extant literature has highlighted that multiple-injured patients are an important component of the injury pattern following earthquakes [37], as, compared to the victims of large-scale disasters, they are at increased risk of suffering from disabilities, they require longer health care and injury-related surgical procedures that often may be difficult to perform immediately after the event [21]. Our analysis revealed that in the L’Aquila earthquake, multiple-injured patients accounted for almost 50% of all admissions. However, 75% of them were admitted to the hospital within the 24 h after the earthquake [29,38]. Moreover, the ratio of females to males in this group was still approximately 2:1, with 66% of the patients being older than 60 years, and 28% with an age range between 17–59 years. The analysis of the median LOS of the multiple trauma patients (Table 9) indicates an average LOS similar to those presenting with single injuries (11.2 days vs. 12.1 days). Furthermore, multiple-injured male patients had a longer average LOS than females (M:12.3 days vs. F:11.2 days); conversely, among the single-injury patients, females had a longer median LOS (M:11.5 days vs. F:12.4 days).

Although fracture-related admissions were the main cause in both periods (2009:46.8% vs. 2010–2016: 45.6%), the comparison with the period 2010–2016 shows that the average LOS for the fracture-related admissions was longer in 2009 (+2.9 days). Moreover, the types of injuries treated by the clinical surgery units after the earthquake had frequencies different from the average rates reported for the period 2010–2016: internal injury of thorax, abdomen, and pelvis (ICD-9: 860–869), +5.1%; intracranial injury (ICD-9: 850–854), +7.2%. These findings, if supported by further studies on the earthquakes occurring in Italy over the past years, have important implications for the regional and national health care system preparedness, in order to ensure the provision of critical public health and medical services during an earthquake, and the adequate clinical and surgical management of the injured (Preparedness).

Notwithstanding the overloading of health care delivery due to the immediate surge of injury-related admissions (Table 1), compounded by the evacuation of the main hospital in L’Aquila, the analysis of the average LOS shows a longer-than-average LOS for surgery patients (12.11 days vs. 10 days). The median stay in hospital decreased progressively within the four days after the earthquake, with men spending an average of 1.5 days less in hospital than women, while average LOSs were longer as patient age increased (13.8 days of average LOS for adults aged older than 60 years compared to the 3.4 days reported for ages 0–16).

The analysis of patient discharge disposition and the comparison with 2010–2016 showed no significant difference between in-hospital mortality percentages (2009:1.8% vs. 2010–2016:1.3%). Discharges to patient’s home (no.2) decreased by 15%, while a rise was observed in the number of interhospital and intrahospital transfers for both acute and rehabilitation care (no.6–8) compared to 2010–2016. This highlights an increased need for health care and rehabilitation in the earthquake-related patients (possibly due to the severity of the injuries) with respect to the patients admitted in the years following the earthquake. Furthermore, a longer-than-average LOS is to be related to psychological trauma and to logistical difficulties due to the collapse or damage of buildings.

## 5. Conclusions

Unfortunately, earthquakes are unpredictable in their timing, location and intensity. Italy is a country prone to seismic activity and, despite the number of studies on the earthquakes both prior to and after 2009, none of them have targeted a systematic analysis of the injuries and of their short- and long-term management. However, the analyses of the injury patterns of the patients admitted to the hospital following the L’Aquila earthquake represent an initial step to understanding the effects and the health care needs during a major natural disaster in Italy.

Earthquake-related injuries should be a rare event in high-income countries, such as Italy, in particular those injuries caused by the collapse of buildings, as occurred in the L’Aquila earthquake. However, we believe that in Italy further studies on injury patterns are needed to enhance emergency preparedness and response of the national health care system. Further research might focus on the analysis of the patients admitted to the rehabilitation and long-term care facilities to establish whether they developed permanent disabilities and whether there had been inequalities in relation to the delivery of and access to health care, in particular for special needs or vulnerable populations. In conclusion, considering the types of injuries observed and the features of the majority of the injured admitted to the hospital following the L’Aquila earthquake, we can state that the study on the injury types and patterns caused by natural or man-made disasters, and their surgical and rehabilitation treatment, is critical to planning special surgical emergency units for the operational management and treatment of specific patient profiles affected by natural disasters, particularly in Italy, a country with a high seismic risk [20].

## Figures and Tables

**Figure 1 ijerph-16-01675-f001:**
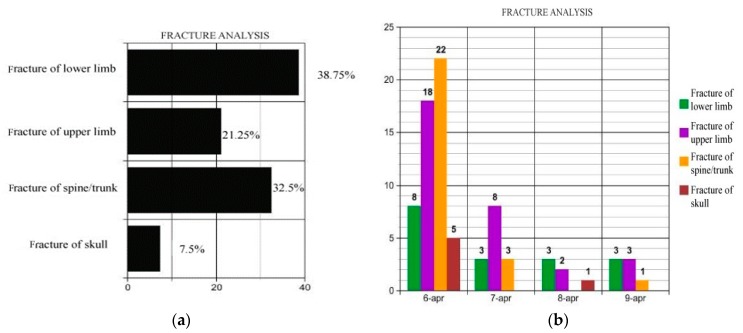
Fractures by anatomic site (**a**) and from 6 to 9 April (**b**).

**Figure 2 ijerph-16-01675-f002:**
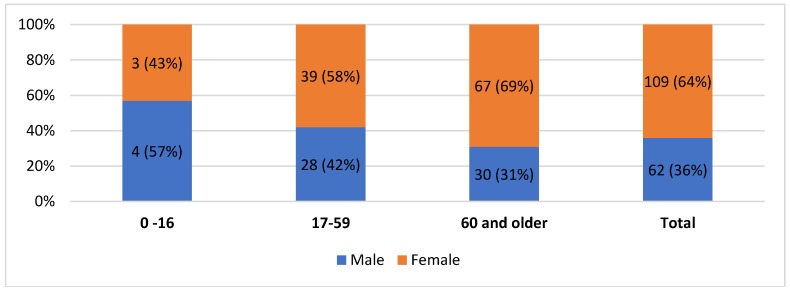
Analysis by age groups and gender of the patients admitted to hospital for traumatic injuries (Crater area, from 6 to 9 April).

**Table 1 ijerph-16-01675-t001:** All-cause hospital admissions among patients from the seismic crater area (ICD-9-CM Chapters).

Hospitalizations by Principal Diagnosis from 6 to 9 April 2009	06/04	07/04	08/04	09/04	Total
Injury and poisoning (800–999)	125 (48.5%)	23 (19.8%)	12 (12.5%)	11 (12.0%)	171 (30.4%)
Diseases of the circulatory system (390–459)	35 (13.6%)	21 (18.1%)	17 (17.7%)	15 (16.3%)	88 (15.7%)
Diseases of the respiratory system (460–519)	13 (5.0%)	14 (12.1%)	7 (7.3%)	9 (9.8%)	43 (7.7%)
Other: Coded V	6 (2.3%)	6 (5.2%)	8 (8.3%)	13 (14.1%)	33 (5.9%)
Neoplasms (140–239)	14 (5.4%)	7 (6.0%)	3 (3.1%)	5 (5.4%)	29 (5.1%)
Diseases of the digestive system (520–579)	12 (4.7%)	7 (6.0%)	6 (6.2%)	4 (4.5%)	29 (5.1%)
Complications of pregnancy, childbirth and of the puerperium (630–677)	14 (5.4%)	2 (1.7%)	7 (7.3%)	6 (6.5%)	29 (5.1%)
Endocrine, nutritional, and metabolic diseases and immunity disorders (240–279)	5 (1.9%)	5 (4.3%)	7 (7.3%)	5 (5.4%)	22 (3.9%)
Diseases of the Musculoskeletal System and Connective Tissue (710–739)	7 (2.7%)	8 (6.9%)	6 (6.2%)	1 (1.1%)	22 (3.9%)
Symptoms, signs, and ill-defined conditions, (780–799)	6 (2.3%)	5 (4.3%)	9 (9.4%)	2 (2.2%)	22 (3.9%)
Mental disorders (290–319)	4 (1.5%)	3 (2.6%)	4 (4.2%)	9 (9.8%)	20 (3.6%)
Diseases of the nervous system and sense organs (320–389)	3 (1.2%)	7 (6.0%)	6 (6.2%)	4 (4.3%)	20 (3.6%)
Diseases of the genitourinary system (580–629)	8 (3.1%)	4 (3.5%)	1 (1.0%)	3 (3.3%)	16 (2.8%)
Disease of the blood and blood-forming organs (280–289)	1 (0.4%)	2 (1.7%)	1 (1.0%)	3 (3.3%)	7 (1.2%)
Diseases of the skin and the subcutaneous tissue (680–709)	3 (1.2%)	-	-	2 (2.2%)	5 (0.9%)
Infectious and parasitic diseases (001–139)	1 (0.4%)	2 (1.7%)	1 (1.0%)	-	4 (0.7%)
Certain conditions originating in the perinatal period (760–779)	1 (0.4%)	-	1 (1.0%)	-	2 (0.3%)
Congenital anomalies (740–759)	-	-	-	-	-
**All ICD-9-CM Chapters**	258 (100%)	116 (100%)	96 (100%)	92 (100%)	562 (100%)

**Table 2 ijerph-16-01675-t002:** Frequencies by date and gender, grouped by injury type (ICD-9-CM, Injury Chapters).

Hospitalizations by Principal Diagnosis from 6 to 9 April 2009	06/04	07/04	08/04	09/04	Male	Female	Ratio M/F	Total
Fractures (800–829)	53	14	6	7	28 (16.4%)	52 (30.4%)	0.53	80 (46.8%)
Internal injury of thorax, abdomen, and pelvis (860–869)	19	3	2	1	10 (5.9%)	15 (8.8%)	0.66	25 (14.6%)
Intracranial injury, excluding those with skull fracture (850–854)	17	2	1	1	11 (6.4%)	10 (5.9%)	1.1	21 (12.3%)
Contusion with intact skin surface (920–924)	8	2	-	-	3 (1.8%)	7 (4.1%)	0.42	10 (5.9%)
Certain traumatic complications and unspecified injuries (958–959)	10	1	2	-	3 (1.8%)	10 (5.9%)	0.3	13 (7.6%)
Late effects of injuries, poisonings, toxic effects, and other external causes (905–909)	5	-	-	-	1 (0.6%)	4 (2.3%)	0.25	5 (2.9%)
Crushing injury (925–929)	4	1	-	-	3 (1.8%)	2 (1.2%)	1.5	5 (2.9%)
Superficial injury (910–919)	2	-	-	-	1 (0.6%)	1 (0.6%)	1	2 (1.2%)
Open wound of lower limb (890–897)	2	-	-	1	-	3 (1.8%)	-	3 (1.2%)
Open wound of head, neck, and trunk (870–879)	2	-	-	-	1 (0.6%)	1 (0.6%)	1	2 (1.2%)
Sprains and strains of joints and adjacent muscles (840–848)	2	-	1	1	1 (0.6%)	3 (1.8%)	0.33	4 (2.3%)
Open wound of upper limb (880–887)	1	-	-	-	-	1 (0.6%)	-	1 (0.6%)
Dislocation (830–839)	-	-	-	-	-	-	-	-
Injury to blood vessels (900–904)	-	-	-	-	-	-	-	-
Effects of foreign body entering through orifice (930–939)	-	-	-	-	-	-	-	-
Burns (940–949)	-	-	-	-	-	-	-	-
Injury to nerves and spinal cord (959–957)	-	-	-	-	-	-	-	-
**All Injuries Chapters**	125 (73.1%)	23 (13.5%)	12 (7%)	11 (6.4%)	62 (36.3%)	109 (63.7%)	0.56	171 (100%)

**Table 3 ijerph-16-01675-t003:** Frequencies sorted by sex among the population in the seismic crater and the overall Abruzzo population admitted to hospital due to an injury in 2009 vs. 2012–2016. (ICD-9-CM, Injury Chapters).

Hospitalizations by Principal Diagnosis from 6 to 9 April	From 6 to 9 April 2009		From 2010 to 2016		
	Male	Female	Male	Female	*p*-Value
Fractures (800–829)	28 (35%)	52 (65%)	23,375 (47%)	26,606 (53%)	0.045 *
Internal injury of thorax, abdomen, and pelvis (860–869)	10 (40%)	15 (60%)	1079 (51%)	1072 (49%)	0.041 *
Intracranial injury, excluding those with skull fracture (850–854)	11 (52%)	10 (48%)	3871 (49%)	3961 (51%)	0.095 (.)
Contusion with intact skin surface (920–924)	3 (30%)	7 (70%)	1329 (49%)	1382 (51%)	n.s.
Certain traumatic complications and unspecified injuries (958–959)	3 (23%)	10 (77%)	1933 (51%)	1844 (49%)	0.080 (.)
Late effects of injuries, poisonings, toxic effects, and other external causes (905–909)	1 (20%)	4 (80%)	5533 (47%)	6162 (53%)	n.s.
Crushing injury (925–929)	3 (60%)	2 (40%)	152 (51%)	144 (49%)	n.s.
Superficial injury (910–919)	1 (50%)	1 (50%)	366 (51%)	351 (49%)	n.s.
Open wound of lower limb (890–897)	-	3 (100%)	358 (53%)	315 (47%)	n.s.
Open wound of head, neck, and trunk (870–879)	1 (50%)	1 (50%)	445 (52%)	403 (48%)	n.s.
Sprains and strains of joints and adjacent muscles (840–848)	1 (25%)	3 (75%)	3064 (51%)	3026 (49%)	n.s.
Open wound of upper limb (880–887)	-	1 (100%)	1699 (52%)	1585 (48%)	n.s.
**All injuries Chapters**	62 (36%)	89 (64%)	43,204 (48%)	46,851 (52%)	0.002 *

(*) statistically significant (*p* < 0.05); (.) marginally significant (0.05 < *p* < 0.010); n.s.: not statistically significant (*p* > 0.10).

**Table 4 ijerph-16-01675-t004:** Frequencies and average length of stay among the population in the seismic crater in 6–9 April 2009 and in the overall Abruzzo region admitted to hospital due to injuries in 2010–2016. (ICD-9-CM, Injury Chapters).

Hospitalizations by Principal Diagnosis from 6 to 9 April	From 6 to 9 April 2009		From 2010 to 2016		
	Total (%)	Average Length of Stay	Total (%)	Average Length of Stay	*p*-Value
Fractures (800–829)	80 (46.8%)	12.8	49,981(45.6%)	9.9	0.08 (.)
Internal injury of thorax, abdomen, and pelvis (860–869)	25 (14.6%)	12.1	21,521 (19.7%)	10.1	n.s.
Intracranial injury, excluding those with skull fracture (850–854)	21 (12.3%)	9.2	7832 (7.2%)	9.1	n.s.
Contusion with intact skin surface (920–924)	10 (5.8%)	7.8	2711 (2.5%)	5.7	n.s.
Certain traumatic complications and unspecified injuries (958–959)	13 (7.6%)	10.6	3777 (3.5%)	4.8	n.s.
Late effects of injuries, poisonings, toxic effects, and other external causes (905–909)	5 (2.9%)	26.8	11,695 (10.7%)	10.61	n.s.
Crushing injury (925–929)	5 (2.9%)	16.8	296 (0.3%)	8.9	n.s.
Superficial injury (910–919)	2 (1.2%)	6	717 (0.7%)	6.5	n.s.
Open wound of lower limb (890–897)	3 (1.7%)	24	673 (0.6%)	5.9	n.s.
Open wound of head, neck, and trunk (870–879)	2 (1.2%)	1.5	848 (0.8%)	5.3	n.s.
Sprains and strains of joints and adjacent muscles (840–848)	4 (2.3%)	3.8	6090 (5.6%)	5	n.s.
Open wound of upper limb (880–887)	1 (0.5%)	17	3284 (3.0%)	3.86	n.s.
**All injuries Chapters**	171 (100%)	12.1	109,425(100%)	9.9	0.03 *

(*) statistically significant (*p* < 0.05); (.) marginally significant (0.05 < *p* < 0.010); n.s.: not statistically significant (*p* > 0.10).

**Table 5 ijerph-16-01675-t005:** Analysis of fracture-related hospitalizations by anatomic site, gender, and average length of stay.

Analysis of Fracture-Related Hospitalizations from 6 to 9 April (ICD-9-CM)	Total	Male	Female	Ratio M/F	Average Length of Stay
**Head injury (800–804)**					
Skull	2 (2.5%)	1 (1.3%)	1 (1.3%)	1	8
Orbital floor	3 (3.8%)	3 (3.7%)	-	-	5
Orbit	1 (1.3%)	1 (1.3%)	-	1	-
**Fracture of the neck and trunk (805–809)**					
Spinal Column	20 (25%)	5 (6.2%)	15 (18.7%)	0.33	18.4
Acetabulum	2 (2.5%)	1 (1.3%)	1 (1.3%)	1	17
Ribs	2 (2.5%)	2 (2.5%)	-	-	2
Pelvic girdle	2 (2.5%)	-	2 (2.5%)	-	34
**Fracture of upper limb (810–819)**					
Humerus	7 (8.8%)	1 (1.3%)	6 (7.5%)	0.16	5.28
Radius and Ulna	8 (10.0%)	2 (2.5%)	6 (7.5%)	0.33	3
Phalanges of hand	1 (1.3%)	-	1 (1.3%)	-	-
Multiple fractures of hand bones	1 (1.3%)	1 (1.3%)	-	-	-
**Fracture of lower limbs (820–829)**					
Femur	15 (18.8%)	3 (3.7%)	12 (15%)	0.25	14.93
Tibia and Fibula	10 (12.5%)	2 (2.5%)	8 (10%)	0.25	8.2
Malleolus	1 (1.3%)	1 (1.3%)	-	-	-
Metatarsal bones	4 (5.0%)	1 (1.3%)	3 (3.7%)	0.33	6.5
Phalanges of foot	1 (1.3%)	1 (1.3%)	-	-	-
**Total**	80 (100%)	25 (31.3%)	55 (68.7%)	0.45	12.8

**Table 6 ijerph-16-01675-t006:** Frequencies sorted by age among the population in the seismic crater and the overall Abruzzo population admitted to hospital due to an injury in 2009 vs. 2010–2016 period. (ICD-9-CM, Injury Chapters).

Hospitalizations by Principal Diagnosis from 6 to 9 April 2009	Period	0–16	17–59	60 and Older	*p*-Value
Fractures (800–829)	6 to 9 April 2009	5 (6.3%)	27 (33.8%)	48 (60%)	n.s.
	2010–2016	3762 (7.9%)	19,092 (39.9%)	24,957 (52.2%)	
Internal injury of thorax, abdomen, and pelvis (860–869)	6 to 9 April 2009	-	11 (44%)	14 (56%)	n.s.
	2010–2016	162 (7.9%)	922 (44.9%)	968 (47.2%)	
Intracranial injury, excluding those with skull fracture (850–854)	6 to 9 April 2009	1 (4.8%)	5 (23.8%)	15 (71.4%)	n.s.
	2010–2016	798 (10.7%)	2673 (36%)	3961 (53.3%)	
Contusion with intact skin surface (920–924)	6 to 9 April 2009	-	5 (50%)	5 (50%)	n.s.
	2010–2016	227 (8.8%)	979 (38%)	1366 (53.2%)	
Certain traumatic complications and unspecified injuries (958–959)	6 to 9 April 2009	-	7 (53.8%)	6 (46.2%)	n.s.
	2010–2016	647 (18.2%)	1408 (39.6%)	1502 (42.2%)	
Late effects of injuries, poisonings, toxic effects, and other external causes (905–909)	6 to 9 April 2009	-	2 (40%)	3 (60%)	n.s.
	2010–2016	812 (7.2%)	4384 (39%)	6057 (53.8%)	
Crushing injury (925–929)	6 to 9 April 2009	-	4 (80%)	1 (20%)	n.s.
	2010–2016	14 (4.9%)	133 (46.2%)	141 (48.9%)	
Superficial injury (910–919)	6 to 9 April 2009	1 (50%)	1 (50%)	-	n.s.
	2010–2016	85 (12.4%)	285 (41.5%)	316 (46.1%)	
Open wound of lower limb (890–897)	6 to 9 April 2009	-	1 (30%)	2 (70%)	n.s.
	2010–2016	52 (8.2%)	271 (42.5%)	314 (49.3%)	
Open wound of head, neck, and trunk (870–879)	6 to 9 April 2009	-	2 (100%)	-	n.s.
	2010–2016	88 (17.4%)	313 (61.7%)	406 (20.9%)	
Sprains and strains of joints and adjacent muscles (840–848)	6 to 9 April 2009	-	2 (50%)	2 (50%)	n.s.
	2010–2016	338 (5.8%)	2741 (46.7%)	2792 (47.5%)	
Open wound of upper limb (880–887)	6 to 9 April 2009	-	-	1 (100%)	n.s.
	2010–2016	249 (7.9%)	1381 (44%)	1510 (48.1%)	
**All injuries Chapters**	6 to 9 April 2009	7 (4.2%)	67 (39.1%)	97 (56.7%)	n.s.
	2010–2016	7560 (8.4%)	35,510 (39.6%)	46,601 (52%)	

n.s.: not statistically significant (*p* > 0.10).

**Table 7 ijerph-16-01675-t007:** Multiple trauma patients by day of admission, age group and gender.

Total Hospitalizations	06/04	07/04	08/04	09/04	Total
	125	23	12	11	171
Hospitalizations of multiple trauma patients (% daily and total)	74(59%)	7 (30%)	5 (41%)	4 (36%)	90 (52%)
Multiple trauma patients: Female (% total)	47	2	5	3	57 (63%)
Multiple trauma patients: Male (% total)	27	5	-	1	33 (37%)
Multiple trauma patients: Age group (% total)					
(0–16)	5	-	-	-	5 (6%)
(17–59)	24	1	-	-	25 (28%)
(60 and older)	45	6	5	4	60 (66%)

**Table 8 ijerph-16-01675-t008:** Comparison of discharge patterns for patients admitted to hospital (total and divided by sex, between 6–9 April 2009 and 2010–2016 period).

All Injuries Discharge Patterns	06/04–09/04			2010–2016
	Male	Female	Total	Total
**1** (Deceased patient)	3 (4.8%)	-	3 (1.8%)	1201 (1.3%)
**2** (Discharge to patient’s home)	44 (69.8%)	77 (70.6%)	121 (70.8%)	77,635 (84.5%)
**3** (Discharge to residential care homes)	3 (4.8%)	7 (6.4%)	10 (5.8%)	1382 (1.5%)
**4** (Discharge to home under care of organized home health service)	1 (1.6 %)	-	1 (0.6%)	1019 (1.1%)
**5** (Left against medical advice)	3 (4.8%)	7 (6.4%)	10 (5.8%)	3784 (4.1%)
**6** (Transfer to another, public or private, acute care facility)	3 (4.8%)	11 (10.1%)	14 (8.2%)	1548 (1.7%)
**7** (Transfer to another type of inpatient care within the same health care facility)	2 (3.2%)	3 (2.8%)	4 (2.3%)	2326 (2.5%)
**8** (Transfer to another public or private inpatient rehabilitation facility)	4 (6.3%)	4 (3.7%)	8 (4.7%)	2989 (3.3%)
	n.s.		<0.001 *	
**All Discharge patterns**	62	109	171 (100%)	91,884 (100%)

(*) statistically significant (*p* < 0.05); n.s.: not statistically significant (*p* > 0.10).

**Table 9 ijerph-16-01675-t009:** Average length of stay (in parentheses) by all-cause hospital admissions, injuries, days, age group and sex.

LOS	06/04	07/04	08/04	09/04	Total
All-cause admissions	258 (13.8)	116 (13)	96 (14.8)	92 (30)	562 (16.6)
All Injuries	125 (13.2)	23 (10.8)	12 (7.5)	11 (7.3)	171 (12.1)
Injuries (Males)	45 (12.2)	12 (10.2)	3 (4.7)	2 (14)	62 (11.5)
Injuries (Females)	80 (13.8)	11 (11.5)	9 (8.4)	9 (5.8)	109 (12.4)
All Injuries Age Group					
(0–16)	-	-	-	-	7 (3.4)
(17–59)	-	-	-	-	67 (10.6)
(60 and older)	-	-	-	-	97 (13.8)

**Table 10 ijerph-16-01675-t010:** Average length of stay of multiple trauma patients.

LOS	Total
	(Average Length of Stay)
Multiple trauma patients	90 (11.2)
Multiple trauma patients (Male)	33 (12.3)
Multiple trauma patients (Female)	57 (11.2)

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
