# Peer review of "Retrospective Analysis of Injuries and Hospitalizations of Patients Following the 2009 Earthquake of L’Aquila City"

_ijerph, 2019, doi:10.3390/ijerph16101675_

Round 1

Reviewer 1 Report

Thank you for the opportunity to review this paper. The paper describes injury patterns after an earthquake in Italy in 2009.

In its current form, I am of the opinion that the paper does not provide any new or novel insights about the injury profiles after an earthquake. Even though it provides specific information for Italy, I think there have been other studies and reviews that describe the injury profiles better. This paper does describe distinct differences between male and female injury patterns and I think exploring this aspect can provide interesting and novel insights.

The methods need more detail and explanation. Why only the admission for the 6-9 April period. This was when the major shock occurred, but from the information in the introduction, there were some other major shocks as well. The comparison with other years need further explanation - why was this done and why only report the comparison with the year 2012? If the authors do want to compare with other data years, they could compare 2009 with the averages for the data years following 2009. It should be noted that the data is now 10 years old. to a

The result section presents much information, but I think the section will benefit from being critically revised. Figure 1 does not contribute to the paper and the information can be provided in the text. As indicated above, I have concerns about the comparison with only 2012 data. I suggest removing the comparison or presenting a comparison with averaged data for 2010-2016.

The paper reports on 171 admissions following the earthquake, but where all of these admissions due to the earthquake? I think this needs to be clarified.

I suggest ordering Table 1 according to the frequency of occurrence in descending order as done in other tables in the paper, rather than according to the sequential order of diagnosis codes.

I also suggest exploring the results in regard to male and female injury patterns and to take age into account as well. A graph that presents male and female presentations by age groups will contribute to understanding. Exploring these aspects will enhance the paper's originality and provide interesting insights.

I suggest including Male to Female ratios in the tables and excluding rows with 0 cases. Having these extra lines does not contribute any information and makes reading more difficult.

Please revise Table 6. The reader needs to spend too much time and effort to decipher this table. The data might be better as a graph.

In Australia, we are always careful to present data in such a way to avoid the identification of individuals. I have concern that presenting the data in Table 9 can lead to people in the know being able to identify where individuals were treated, etc. Also, the information in Table 9 is not meaningful for people outside of Italy as we do not have an understanding of these facilities. I suggest excluding the table or grouping facilities according to their size, functions, etc. in order to make the information understandable for an international audience.

My concerns about being able to identify individuals need to be taken into account in regard to other tables and data presented.

I think the discussion need to include considerations about why there were the differences between males and females and the age groups seen. Do these profiles reflect the population distribution of the region? Why did men have a certain profile and women another?

Overall, I think the paper will benefit from a more structured presentation of the results and a focus on exploring the gender and age differences seen in the data.

Author Response

Response to Reviewer 1 Comments

Dear Sir,

I would like to thank you for your precious advices. These are our point-by-point response to your comments.

Point 1: In its current form, I am of the opinion that the paper does not provide any new or novel insights about the injury profiles after an earthquake. Even though it provides specific information for Italy, I think there have been other studies and reviews that describe the injury profiles better. This paper does describe distinct differences between male and female injury patterns and I think exploring this aspect can provide interesting and novel insights.

Response 1: We decided to conduct a further statistical analysis between male and female injury patterns.

Point 2: The methods need more detail and explanation. Why only the admission for the 6-9 April period. This was when the major shock occurred, but from the information in the introduction, there were some other major shocks as well. The comparison with other years need further explanation - why was this done and why only report the comparison with the year 2012? If the authors do want to compare with other data years, they could compare 2009 with the averages for the data years following 2009. It should be noted that the data is now 10 years old. 

Response 2:  We also compared 2009 vs the averaged 2010-2016 period as suggested. we provide more accurate Methods. We considered admission for April 6-9 because it has been shown that the 4 days following the main shock are essential to save the injured. We made a comparison with the averaged 2010-2016 period as suggested (Table 3a, Table 3b, Table 5, Table 7)

Point 3: The result section presents much information, but I think the section will benefit from being critically revised. Figure 1 does not contribute to the paper and the information can be provided in the text. As indicated above, I have concerns about the comparison with only 2012 data. I suggest removing the comparison or presenting a comparison with averaged data for 2010-2016.

Response 3: We revised the result section as suggested. We eliminate Figure 1. As written above, we made a comparison with averaged 2010-2016 period.

Point 4: The paper reports on 171 admissions following the earthquake, but where all of these admissions due to the earthquake? I think this needs to be clarified.

Response 4: They were all admitted due to the earthquake. We considered the hospitalisation after the 03.32 AM of the 6 april (time of the main shock). We also excluded from the study outpatient surgeries, rehabilitation hospital stays, and long-stay hospital patients.

Point 5: I suggest ordering Table 1 according to the frequency of occurrence in descending order as done in other tables in the paper, rather than according to the sequential order of diagnosis codes.

Response 5: We’ve done it.

Point 6: also suggest exploring the results in regard to male and female injury patterns and to take age into account as well. A graph that presents male and female presentations by age groups will contribute to understanding. Exploring these aspects will enhance the paper's originality and provide interesting insights.

Response 6: We made further inferential analysis regarding male and female injury patterns. We also highlight a new aspect of age groups at Table 5

Point 7: I suggest including Male to Female ratios in the tables and excluding rows with 0 cases. Having these extra lines does not contribute any information and makes reading more difficult.

Response 7: we’ve done it.

Point 8: Please revise Table 6. The reader needs to spend too much time and effort to decipher this table. The data might be better as a graph.

Response 8: “Table 6” is now included in “Table 7” due to the make the data more readable.

Point 9: In Australia, we are always careful to present data in such a way to avoid the identification of individuals. I have concern that presenting the data in Table 9 can lead to people in the know being able to identify where individuals were treated, etc. Also, the information in Table 9 is not meaningful for people outside of Italy as we do not have an understanding of these facilities. I suggest excluding the table or grouping facilities according to their size, functions, etc. in order to make the information understandable for an international audience.
My concerns about being able to identify individuals need to be taken into account in regard to other tables and data presented.

Response 9:  Thank you for your advice. We decided to eliminate Table 9.

Point 10: I think the discussion need to include considerations about why there were the differences between males and females and the age groups seen. Do these profiles reflect the population distribution of the region? Why did men have a certain profile and women another?

Response 10: We analysed the population distribution between Abruzzo region and all injured patients from 6-9 April 2009; at lines 397-402.

Point 11: Overall, I think the paper will benefit from a more structured presentation of the results and a focus on exploring the gender and age differences seen in the data.

Response 11: We have more structured the presentation of the results while exploring the gender and age differences.

Thank you for your help in this matter. Yours faithfully,

Dr Jacopo Del Papa

Reviewer 2 Report

Dear authors,

Thank you for an interesting paper. I have some suggestions to further improve your paper:

·         Materials and Methods section

Is it correctly understood that the total number of included patients is 171? Please, clarify the inclusion process. Maybe, a chart or figure that showed the sampling process would help?

·         Statistics (line 116 and forward)

The statistical tests used should be mentioned here.

·         Table 1

I suggests that you put all texts in the first columns to either the right or left in the table, in order to easy the reading.

·         Table 2

I suggests that you put all texts in the first columns to either the right or left in the table, in order to easy the reading. Also, I don´t understand the numbers behind the first line, for example Fractures (800-829), I think it might be the diagnoses codes? Please explain this information in the table text, so that you can understand the tables exclusively, without any other information.

·         Comparison of the injury-related hospitalisations between 6-9 April 2009 and 2012 Table 3.

Did you do any statistical tests to determine if the differences are significant or not? Otherwise, I suggest you to do so.

·         Line 353-256

This is a methodology section part, not a result.

·         Table 5 and 6

Is the numbers presented nb of persons? And than the %? I suggest a clarification in the table, and also that the % is only presented as one number, not using two decimals. It would make the table more easily read.  

·         Table 7

Would it be possible to write in what discharge patters nb is? It seems to me that you have planet of space left in the table to do so, and it would make it much more easy to understand. A table should be possible to read and understand without any other text.

·         Distribution of admissions across regional and extra-regional hospitals and Table 9

This is maybe very informative if you are familiar with Italian and specifically the regional geography, but for me, this sections only say that patients were spread in many hospitals. Could you maybe add a map, or some information on what kind of hospitals that were admitting patients, their geographical position in relation to the epicenter os something that put these numbers in a context?

·         Conclusions

I suggest a more concise and distinct conclusion; what did your study show?

In line 418-421,  you say that the fact that few patients were admitted in out of region hospitals shows that the health cares system were adequate. This is not fully understandable for me as reader, when it does not say anything on if the patients were admitted to the CORRECT level of care, anywhere. Could there be any other explanations, such as limited transport resources, or health care system factors?

In line 404-407, you discuss that the long term needs are related to rehabilitation and psychological support, but in line 429, you only focus on surgical teams and units.

I also wonder if no patients died after being admitted to hospital?

I miss a discussion of methodological limitations for this study.

Author Response

Response to Reviewer 2 Comments

Dear Sir,

I would like to thank you for your precious advices. These are our point-by-point response to your comments:

Point 1: Is it correctly understood that the total number of included patients is 171? Please, clarify the inclusion process. Maybe, a chart or figure that showed the sampling process would help?

Response 1: Yes, they are 171. We take all the patients admitted due to the earthquake. We considered the hospitalitation after the 03.32 AM of the 6 april (time of the main shock). We also excluded from the study outpatient surgeries, rehabilitation hospital stays, and long-stay hospital patients.

Point 2: Statistics (line 116 and forward). The statistical tests used should be mentioned here.

Response 2: Thank you for the advice. We have mentioned the statistical test.

Point 3: Table 1. I suggest that you put all texts in the first columns to either the right or left in the table, in order to easy the reading.

Response 3: We’ve done it.

Point 4: Table 2. I suggest that you put all texts in the first columns to either the right or left in the table, in order to easy the reading. Also, I don´t understand the numbers behind the first line, for example Fractures (800-829), I think it might be the diagnoses codes? Please explain this information in the table text, so that you can understand the tables exclusively, without any other information.

Response 4: We’ve done it.

Point 5: Comparison of the injury-related hospitalisations between 6-9 April 2009 and 2012 Table 3. Did you do any statistical tests to determine if the differences are significant or not? Otherwise, I suggest you to do so.

Response 5: We decided to make a comparison between 2009 and the average of 2010-2016. We also did a chi-test and fisher test to see if there are significant differences.

Point 6: Line 253-256.This is a methodology section part, not a result.

Response 6: We decided to delete that period.

Point 7: Table 5 and 6. Is the numbers presented nb of persons? And than the %? I suggest a clarification in the table, and also that the % is only presented as one number, not using two decimals. It would make the table more easily read.  

Response 7: “Table 5” is now “Table 6” and we made the table more easily to read. “Table 6” is now included in “Table 7” due to make the data more readable.

Point 8:  Table 7. Would it be possible to write in what discharge patters nb is? It seems to me that you have planet of space left in the table to do so, and it would make it much more easy to understand. A table should be possible to read and understand without any other text.

Response 8: As above, “Table 6” is now included in “Table 7” due to make the data easier to understand.

Point 9:  Distribution of admissions across regional and extra-regional hospitals and Table 9. This is maybe very informative if you are familiar with Italian and specifically the regional geography, but for me, this sections only say that patients were spread in many hospitals. Could you maybe add a map, or some information on what kind of hospitals that were admitting patients, their geographical position in relation to the epicenter os something that put these numbers in a context?

Response 9: Thank you for the advice. We decided to eliminate Table 9 because it's not so relevant to the comprehension of our study.

Point 10:  Conclusions. I suggest a more concise and distinct conclusion; what did your study show?
In line 418-421, you say that the fact that few patients were admitted in out of region hospitals shows that the health cares system were adequate. This is not fully understandable for me as reader, when it does not say anything on if the patients were admitted to the CORRECT level of care, anywhere. Could there be any other explanations, such as limited transport resources, or health care system factors?
In line 404-407, you discuss that the long term needs are related to rehabilitation and psychological support, but in line 429, you only focus on surgical teams and units.

 I also wonder if no patients died after being admitted to hospital?

 I miss a discussion of methodological limitations for this study.

Response 10: We eliminate line 418-421. In table 7, the discharge pattern N.1 (deceased patient) is the mortality during hospitalisations. Methodological Limitations are in lines 137-142.

Thank you for your help in this matter. Yours faithfully,

Dr Jacopo Del Papa

Round 2

Reviewer 1 Report

Page 3, line 118: Please write out HDR the first time it is used. The explanation is given in line 138 on the same page, but please define in line 118.

Please make a statement regarding whether all injuries were due to the earthquake. I assume they were and without external cause codes will be difficult to ascertain otherwise.

I suggest adding a column with M:F ratios for Tables 2 and 4.

Please capitalise the axis titles of Figure 1a.

I suggest changing Figure 2 and 3 to show a gender by age distribution. I am interested in knowing if there were age differences by gender. This might be part of the explanation of the high proportion of female injuries.

Table 5 is difficult to interpret. I suggest a bar graph/horizontal bar graph.

Please capitalise the description in Table 7, i.e., Deceased person ...

Please elaborate on what is meant with "a different approach in dealing with a sudden catastrophic event between the male and female". Does age play a role here?

Author Response

Response to Reviewer 1 Comments (Second Round)

Dear Sir,

I would like to thank you for your precious advices. These are our point-by-point response to your comments.

Point 1:
Page 3, line 118: Please write out HDR the first time it is used. The explanation is given in line 138 on the same page, but please define in line 118. 
Response 1: Ok, we’ve done it.

Point 2: Please make a statement regarding whether all injuries were due to the earthquake. I assume they were and without external cause codes will be difficult to ascertain otherwise.
Response 2: We decided to make a statement at lines 145-150.

Point 3:
I suggest adding a column with M:F ratios for Tables 2 and 4.
Response 3: OK, we’ve done it.

Point 4:
Please capitalise the axis titles of Figure 1a.
Response 4: OK We’ve done it.

Point 5:
I suggest changing Figure 2 and 3 to show a gender by age distribution. I am interested in knowing if there were age differences by gender. This might be part of the explanation of the high proportion of female injuries.
Response 5: We decided to combine “Figure 2” and “Figure 3” in the new “Figure 2” to show if there were age differences by gender.

Point 6: Table 5 is difficult to interpret. I suggest a bar graph/horizontal bar graph. 
Response 6: Thank you for the advice, we decided to change the table in a better way.

Point 7: Please capitalise the description in Table 7, i.e., Deceased person ... 
Response 7: Ok, we’ve done it.

Point 8: Please elaborate on what is meant with "a different approach in dealing with a sudden catastrophic event between the male and female". Does age play a role here?
Response 8: The sentence was a bit short and unclear for the broad context of gender and disaster management. Therefore, we limited our discussion on commenting our findings (i.e., that age may have played a role in the observed gender differences), we mentioned other possible aspects that the literature highlights as possible factors, and we added three citations that may help a reader to further delve into such a topic (Lines 406-408) .

Thank you for your help in this matter. Yours faithfully,

Dr Jacopo Del Papa

Reviewer 2 Report

Dear authors, 

Thank you for resubmitting your paper. I think the paper is much more clear and consist in this version, and especially the tables is more easily read. 

However, I still have two suggestion; 

1) Line 422 and forward. You present some interesting differences in inuries and hospital stays based on gender. But you don´t discuss them. Whar are possible cases for these findings? Has it been analysed before? Is the difference significant in all ages? I suggest some more discussion on this, in order to highlight an interesting result of your study. 

2) I suggest you to consider if the title highlight the main finding os your study. In what way is magagement analyzed, otherwise than time for hospitalization? 

Otherwise, I´m plased to read all improvments and wish you good luck!

Author Response

Response to Reviewer 2 Comments (Second Round)

Dear Sir,

I would like to thank you for your precious advices. These are our point-by-point response to your comments.

Point 1: Line 422 and forward. You present some interesting differences in injuries and hospital stays based on gender. But you don´t discuss them. What are the possible causes for these findings? Has it been analysed before? Is the difference significant in all ages? I suggest some more discussion on this, in order to highlight an interesting result of your study. 

Response 1: Thank you for your precious collaboration. We decided to focus on gender and age groups, changing “Figure 2” (with a statistical test in lines 299-300). We make another comparison in lines 406-408.

Point 2: I suggest you consider if the title highlights the main finding of your study. In what way is management analyzed, otherwise than time for hospitalization? 
Response 2: We decided to change “management” in “hospitalisation” in the title.

Thank you for your help in this matter. Yours faithfully,

Dr Jacopo Del Papa